# Cotton and Flax Textiles Leachables Impact Differently Cutaneous *Staphylococcus aureus* and *Staphylococcus epidermidis* Biofilm Formation and Cytotoxicity

**DOI:** 10.3390/life12040535

**Published:** 2022-04-06

**Authors:** Chloé Catovic, Imen Abbes, Magalie Barreau, Catherine Sauvage, Jacques Follet, Cécile Duclairoir-Poc, Anne Groboillot, Sandra Leblanc, Pascal Svinareff, Sylvie Chevalier, Marc G. J. Feuilloley

**Affiliations:** 1Unité de Recherche Communication Bactérienne et Stratégies Anti-Infectieuses (UR4312 CBSA), Université de Rouen-Normandie, F-27000 Evreux, France; chloe.catovic1@univ-rouen.fr (C.C.); abbes_imen@hotmail.fr (I.A.); magalie.barreau@univ-rouen.fr (M.B.); cecile.duclairoir-poc@univ-rouen.fr (C.D.-P.); anne.groboillot@univ-rouen.fr (A.G.); sylvie.chevalier@univ-rouen.fr (S.C.); 2Association Lin et Chanvre Bio, F-76450 Saint Vaast Dieppedalle, France; catherine.sauvage@orange.fr (S.C.); jacfollet@hotmail.fr (J.F.); 3Biogalenys SAS, F-27930 Miserey, France; sandra.leblanc@biogalenys.com (S.L.); pascal.svinareff@biogalenys.com (P.S.)

**Keywords:** skin bacteria, textiles leachables, *Staphylococcus aureus*, *Staphylococcus epidermidis*, growth, biofilm, biosurfactants, surface polarity, resistance to antibiotics, cytotoxicity

## Abstract

Bacteria can bind on clothes, but the impacts of textiles leachables on cutaneous bacteria remain unknown. Here, we studied for the first time the effects of cotton and flax obtained through classical and soft ecological agriculture on the representatives *S. aureus* and *S. epidermidis* bacteria of the cutaneous microbiota. Crude flax showed an inhibitory potential on *S. epidermidis* bacterial lawns whereas cotton had no effect. Textile fiber leachables were produced in bacterial culture media, and these extracts were tested on *S. aureus* and *S. epidermidis*. Bacterial growth was not impacted, but investigation by the crystal violet technique and confocal microscopy showed that all extracts affected biofilm formation by the two staphylococci species. An influence of cotton and flax culture conditions was clearly observed. Flax extracts had strong inhibitory impacts and induced the formation of mushroom-like defense structures by *S. aureus.* Conversely, production of biosurfactant by bacteria and their surface properties were not modified. Resistance to antibiotics also remained unchanged. All textile extracts, and particularly soft organic flax, showed strong inhibitory effects on *S. aureus* and *S. epidermidis* cytotoxicity on HaCaT keratinocytes. Analysis of flax leachables showed the presence of benzyl alcohol that could partly explain the effects of flax extracts.

## 1. Introduction

Because of its role in skin homeostasis, aging, and welfare, the cutaneous microbiota is now as a major industrial and societal center of interest [1]. It is now almost impossible to launch new cosmetic products without claims on the respect of the cutaneous microbiota, when this microbiota is not used itself as a target or a tool for the final product [2,3]. By its size and diversity, the cutaneous microbiota is the second of the human body after the intestinal microbiota and includes bacteria, viruses, yeast, fungi, and archaea. In adults, the bacterial microbiota includes three major phyla, *Actinobacteria*, *Firmicutes* and *Proteobacteria*. The more representative species are *Corynebacterineae*, *Propionibacterineae*, *Microococcineae,* and *Staphylococcaceae* [4,5]. This microbiota is in constant interaction with the skin and represents the first barrier of the human body against environmental factors such as UV and pollutants [6,7,8,9]. Bacteria colonize not only the surface but also the depths of the stratum corneum where they associate preferentially with sweat ducts, hair follicles, and furrows [10]. Then, skin-associated bacteria are largely diffused in sweat, and some genera including *Corynebacterium*, *Staphylococcus,* and *Anaerococcus* are at the origin of body odor formation through sweat molecule metabolization [11,12,13]. Humidity and friction favor the transfer of bacteria to textiles [14]. Some species such as *Staphylococcus aureus* can survive on cotton or polyester for up to 3 weeks [15]. The nature and composition of textiles are determinant parameters controlling colonization by skin microorganisms [16]. Bacteria adhere preferentially to artificial fibers such as polyamide and polyester and much less to natural fibers such as cotton [17,18]. This process is also modulated by the effect of textiles on the cutaneous microenvironment which directly affects skin physiology and sweat production [19] as well as cutaneous microflora composition [20]. Bacterial adhesion to textile fibers has multiple consequences including surface alteration, biofilm formation, and biodegradation [21,22]. Textiles and clothes can even favor skin colonization by environmental opportunistic pathogens [23].

Whereas the association of microorganisms to textiles has been a major center of interest [18], the effect of textile fibers on cutaneous bacteria remains almost non-documented. Some natural fibers, such as flax, have been used since antiquity for their positive influence on skin [24]. Flax seed oil has a broad antibacterial spectrum against pathogens [25], but the direct impact of flax fabric and clothes on cutaneous bacteria remains to be studied. Originally, the unique study on the effect of clothes on the cutaneous microbiota was realized on specific high technology textiles employed by astronauts. This work did not show any significant variation of the microbiome distribution and composition [26]. However, as all materials, textiles release leachables with potential toxic and stress-inducing potential [27]. It was even demonstrated that textile fibers are keeping traces of the culture conditions of plants from which they were produced, including pesticides and fertilizers [28]. The cutaneous microbiota can evolve under the influence of its microenvironment not only in term of composition but also in regard of its level of virulence. The intrinsic virulence, biofilm formation activity, and resistance to antibiotics of cutaneous microorganisms such as *Staphylococcus epidermidis*, *Staphylococcus aureus,* or *Cutibacterium* (former *Propionibacterium*) *acnes* can be affected by natural skin molecules such as substance P [29], calcitonin gene related peptide [30], and natriuretic peptides [31], and also exogenous factors such as cosmetics [32,33] or air pollutants [8,34,35]. Then, even without acting on their growth, textiles can directly modify cutaneous microbiota virulence and alter skin homeostasis.

In the present study, we investigated the impact of cotton and flax obtained through classical and soft ecological agriculture on two representative species of the cutaneous microbiota, *S. aureus* and *S. epidermidis*. The direct effect of textiles was assayed on the two bacteria, and extracts were produced to investigate the potential of textile leachables on the growth kinetic, biofilm formation activity, biosurfactant production, surface properties, resistance to antibiotics, cytotoxicity, and inflammatory potential of both microorganisms. Chemical analysis of the extracts suggested a potential explanation of observed effects.

## 2. Materials and Methods

### 2.1. Textiles Samples

Four types of textiles were tested: on one side, classical industrial cotton (CIC) and classical industrial flax (CIF) produced from genetically modified (GMO) crops using pesticides and artificial fertilizers, and on the other side, soft organic cotton (SOC) and soft organic flax (SOF) produced from non-GMO crops in the absence of chemical pesticides or fertilizers. All textile samples were provided by Lin et Chanvre Bio (Saint Vaast Dieppedalle, France). These textiles were selected for their production and weaving in the absence of any treatment (chloride, dye, biocide, flame retardant…). Textile samples were cut in small pieces (1 × 1 cm for direct contact studies and 1 × 2.5 cm for extracts production). Except samples used for testing by direct contact with bacteria, all samples were sterilized by autoclaving at 121 °C for 30 min in an airtight container to protect from humidity. For preparation of the extracts, fabric samples were incubated for 24 h at 37 °C in a bacterial growth medium (reinforced *Clostridium* medium, RCM) as proposed by the regulation for preparation of leachables in content–container interaction studies [36]. Fragments and fibers were removed by centrifugation (1500× *g*; 5 min) and filtration through 0.2 µm acetate filters.

### 2.2. Bacterial Strains and Culture Conditions 

*S. aureus* MFP03 and *S. epidermidis* MFP04 were isolated from the skin of healthy volunteers [37]. These bacteria were initially characterized on API^®^ strips and by 16S ribosomal subunit DNA gene sequencing and MALDI-Biotyper^®^ whole proteome analysis. They were subsequently submitted to complete genome sequencing [38]. Strains were stored on biobeads in cryofreezer at −140 °C. Before use, biobeads were transferred into RCM and incubated overnight at 37 °C. Bacteria were then sub-cultured at 37 °C in normal RCM or textile extracts in RCM produced as described in Section 2.1. Experiments were realized using bacteria reaching the middle of the exponential growth phase.

### 2.3. Testing by Textiles Direct Contact

For these tests, Luria-Bertani (LB)-agar Petri dishes were seeded homogeneously in surface by spreading with beads 200 μL of bacterial inoculum (1 × 10^7^ CFU/mL). When the surface dried, sterilized or not, 1 cm^2^ textiles samples were deposited at the center of the dishes. The plates were incubated at 37 °C for 72 h. Every 24 h, Petri dishes were observed and scanned. These experiments were carried out in at least three replicates.

### 2.4. Bacterial Growth Kinetics

For monitoring of the bacterial growth, an overnight bacterial culture in RCM medium was diluted at OD_580nm_ = 0.08 in fresh RCM or CIC, SOC, CIF, or SOF extracts in RCM. Immediately after, 200 µL of each bacterial suspension was dropped in sterile 100-wells flat-bottomed plastic culture plates (Honeycomb, Bioscreen, Helsinki, Finland). The plates were incubated for 24 h at 37 °C under constant agitation in a Bioscreen microplate reader. The OD of each well was measured at 580 nm every 15 min. Five wells were used for every sample, and experiments were carried out in at least three replicates.

### 2.5. Measurement of Biofilm Formation Activity by Crystal Violet Staining

Biofilms formation was studied in 24-wells flat-bottom polystyrene plates (NUNC, Fisher scientific, Roskilde, Denmark) following a procedure adapted from O’Toole [39]. Bacteria from overnight pre-cultures in RCM (mid exponential growth phase) were collected by centrifugation (7500× *g*; 10 min) and rinsed with sterile physiological water (NaCl 0.9%). One milliliter aliquots of bacterial culture adjusted to OD_580nm_ = 0.1 (corresponding to 5.10^7^ CFU/mL) were layered in plates and incubated for 2 h at 37 °C to allow primary adhesion. Then, physiological water was removed from the wells, and RCM or textile extracts in RCM were added. The plates were incubated for 22 h in static condition at 37 °C. At the end of the incubation period, wells were washed two times with sterile physiological water to remove remaining planktonic bacteria. The plates were dried and stained with 0.1% crystal violet for 10 min at room temperature. Afterwards, washes with physiological saline were repeated until the rinsing water became clear. The wells were dried, and 100 µL of ethanol (80% in water) was added. When the biofilm was completely dissolved, the solution was collected and homogenized. The OD_570nm_ of the solution was measured over a linear range from 0.1 to 0.9 using a Thermo Scientific Genesys 20 spectrophotometer (Thermo Fisher Scientific, Illkirch-Graffenstaden, France). Results were expressed as percentage of the control values. Two wells were used for every sample, and experiments were carried out in at least three independent replicates.

### 2.6. Confocal Laser Scanning Microscopy 

Biofilms were formed in the same conditions as for crystal violet studies using specific thin flat glass 24-wells plates (Sensoplate, Greiner bio-One, Kremsmünster, Austria). 

For all biofilms, at the end of the incubation period, wells were washed with sterile physiological water to remove remaining planktonic bacteria and were stained with 300 µL of 5 µM Syto 9 Green Fluorescent Nucleic Acid Stain (Fisherscientific, Invitrogen, Waltham, USA) prepared in sterile physiological water. This staining step was realized at room temperature for 30 min in the dark. Then, biofilms were fixed with ProLong Diamond Antifade Mountant (Molecular Probes R). They were observed using a Zeiss LSM710 confocal laser scanning microscope (CLSM Carl Zeiss Microscopy, Oberkochen, Germany) equipped with a 63 × oil immersion objective. Syto 9 was excited at 488 nm, and fluorescence emission was detected using a band pass filter 481–587 nm.

Images were taken every micrometer throughout the whole biofilm depth. For visualization and processing of three-dimensional (3D) images, Zen 2.1 SP1 software (Carl Zeiss Microscopy, Oberkochen, Germany) was used. Quantitative analyses of image stacks were performed using COMSTAT software. Biomass volume (μm^3^/μm^2^) and maximal and average thickness (μm) were determined using ImageJ software (National Institutes of Health, Bethesda, MD, USA). Two wells were used for every sample, and experiments were carried out in at least three replicates.

Mushroom-like structures were studied by triple straining using Syto 9 Green Fluorescent Nucleic Acid Stain (Fisherscientific, Invitrogen, Waltham, MA, USA), Sypro Ruby (Fisherscientific, Invitrogen, Waltham, USA) and CalcoFluor White (Fluorescent Brightener 28, Sigma-Aldrich, Darmstadt, Germany) to visualize bacteria, matrix proteins, and matrix polysaccharides, respectively. In these studies, wells were not washed between each treatment to avoid biofilm destabilization and because planktonic bacteria were eliminated during the different staining steps. Then, at the end of the incubation period and immediately after removal of the culture medium by gentle aspiration, 300 µL of 200 µg/mL CalcoFluor White was added for 30 min. This solution was removed and replaced by 300 µL of Sypro Ruby Protein Gel Stain. Subsequently, 300 µL of 5 µM Syto 9 was added. All incubations were realized for 30 min in the dark. After staining, samples were fixed with ProLong Diamond. Calcofluor, Syto 9 and Sypro Ruby were excited at 405, 488, and 633 nm, respectively. Fluorescence emission was detected between 399 and 479 nm for Calcofluor, 493, at 575 nm for Syto 9, and between 607 and 797 nm for Sypro Ruby. 

### 2.7. Evaluation of Bacterial Biosurfactant Production

In order to investigate the potential production of surfactant by the two strains, the surface tension of the growth media collected after culture in RCM or fabric extracts for 24 h at 37 °C was measured by the pendant drop method using a DSA30 controlled temperature tensiometer equipped with a video camera (Kruss, Hamburg, Germany). After removal of bacterial cell bodies by centrifugation (10 min, 20 °C, 7500× *g*), the supernatant was collected in a syringe that was inserted in the tensiometer. A pressure was exerted on the piston so that a drop of liquid formed in front of the camera. The drop shape was measured at the limit before detachment. The surface tension was obtained from the drop shape analysis software of the tensiometer using the Young-Laplace equation [40]. Volvic water, considered as the reference for its constant surface tension value (72 mN.m^−1^), was used to validate the measures. RCM, CIC, SOC, CIF, and SOF extracts in RCM were also tested as internal controls. These experiments were carried out in at least three replicates.

### 2.8. Characterization of the Bacterial Surface Polarity

The surface polarity of bacteria was determined using the microbial adhesion to solvents (MATS) technique [41]. This technique is based on the measurement of the affinity of bacteria to polar (chloroform and ethyl acetate) and apolar (hexadecane and decane) solvents. Briefly, in each case, 1.2 mL of bacteria resuspended in sterile physiological water at OD_400nm_ = 0.8 (corresponding to 4 × 10^8^ and 4.8 × 10^8^ CFU/mL for *S. aureus* and *S. epidermidis*, respectively) was mixed for 60 s with 0.2 mL of solvent. After separation of the two phases, bacteria were distributed between water and immiscible solvents following their surface properties. Their affinity for each phase was determined by measurement of water OD_400nm_. Affinity of bacteria to hexadecane (apolar solvent) was selected as representative of the bacterial surface hydrophobicity level. Lewis basicity and Lewis acidity were given by the differences of affinity between apolar and monopolar solvents couples (chloroform–hexadecane and ethyl acetate–decane, respectively). Three tubes were used for every sample and experiments were carried out in at least five replicates.

### 2.9. Antibiotics Sensitivity Assay

The potential effect of fabric extracts on *S. aureus* and *S. epidermidis* sensitivity to antibiotics was studied using the disk diffusion method as described in EUCAST (European Committee on Antimicrobial Susceptibility Testing) [42]. Inoculation of the agar Mueller-Hinton medium was carried out in 3 steps using a sterile swab that was immersed in a suspension of bacterial physiological water at 0.5 McFarland. When the surface of the plate dried, nine disks (Oxoid, Fisher Scientific, Santa Fe, USA) were placed on the inoculated agar surface of a 90 mm diameter Petri dish. A total of twenty-four antibiotics were tested against the two bacteria at the concentration defined by EUCAST (Table 1). Plates were incubated at 37 °C for 22 h. The inhibition zone surrounding each antibiotic disk was measured to the nearest millimeter using a ruler. These experiments were carried out in at least three replicates.

### 2.10. Assessment of Bacterial Cytotoxicity and Inflammatory Activities

The cytotoxicity of bacteria and their supernatants was studied in the HaCaT human keratinocyte cell line. These cells, provided by Cell Line Services (Eppelhein, Germany), were grown in 24-wells plates at 37 °C in 5% CO_2_ atmosphere, in Dulbecco’s modified Eagle’s Medium (DMEM, Lonza, Leva, France) containing 25 mM glucose, 10% fetal bovine serum (FBS, Panbiotech, Aidenbach Germany), 2 mM L-glutamine (Lonza, France), and 1% antibiotic mix (penicillin 100 IU/mL and streptomycin 100 μg/mL, HyClone Thermo Scientific, Illkirch-Graffenstaden, France). When the cells reached 80% of confluence, the medium was removed, cells were washed twice with phosphate buffer saline (PBS Lonza, Levallois-Perret, France), and medium without FBS and antibiotics was added during 8 h. Bacterial cultures were centrifugated for 10 min at 7500× *g*, supernatants were collected, and bacteria were diluted in fresh DMEM (*v*/*v*:1/9). Bacteria were resuspended and rinsed again with DMEM. The cells were incubated for 2 h with bacteria at a multiplicity of infection (MOI) 10:1 or bacterial culture supernatant at the same *v*/*v* ratio. Controls were realized by addition of normal RCM or CIC, SOC, CIF, or SOF extracts diluted in DMEM in the same proportions. The amount of lactate dehydrogenase (LDH) released by HaCaT cells following membrane disruption was determined using the Cytotox 96 enzymatic assay (Promega, France) as described by Picot et al. [43]. Two wells were used for every sample, and experiments were carried out in at least three replicates.

The inflammatory response of HaCaT cells to control and fabric-extracts-treated bacteria was evaluated by assaying interleukin 8 (IL8) secretion. HaCaT cells were exposed to bacteria or their supernatants as previously described in the “cytotoxicity studies” sub-section. The amount of IL8 released by HaCaT cells was determined using the Human IL-8 ELISA Kit (KHC0081) (Fisherscientific, Invitrogen, Waltham, USA). Two wells were used for every sample, and experiments were carried out in at least three replicates.

### 2.11. Chemical Analysis of Fabric Extracts 

In order to understand the differences between classical industrial and soft organic fabrics, cotton and flax extracts were analyzed by high performance liquid chromatography (HPLC) coupled to ultra-violet detection (Thermo Fisher Scientific U3000RS HPLC pump equipped with UV-DAD detector set at 220 nm). For that, 1 mL of each sample was submitted to extraction using 49 mL methanol/water (50/50 *v*/*v*) with ortho-phosphoric acid 0.085% (Sigma-Aldrich, Darmstadt, Germany). After 2 h incubation under constant agitation (300 rpm), the solution was filtrated on 0.45 µm regenerated cellulose syringe filters (Thermo Fisher Scientific, Illkirch-Graffenstaden, France). Separation was realized by gradient elution using methanol/water with 0.085% ortho-phosphoric acid (from 5/95 to 75/25 *v*/*v*) and a RP NUCLEODUR C18 Isis 5 µm 250 × 4 mm Machery Nagel column. A molecule of interest was further identified by gas chromatograph coupled to mass spectrometry using a HP Hewlett Packard Agilent 6890 Plus GC System Gas Chromatograph with HP 5973 MSD Mass Selective Detector.

### 2.12. Statistical Analysis

Means with standard error of the mean (SEM) were calculated and plotted. 

The non-parametric Mann–Whitney test was used to compare the means within the same set of experiments when they were not normally distributed (Gaussian). When they were normally distributed, the Student’s *t* test was used. *P* values were calculated with Past 3.x software.

## 3. Results

### 3.1. Effects of Crude and Sterilized Cotton and Flax Textiles on Staphylococcus aureus and Staphylococcus epidermidis Bacterial Lawns in Petri Dishes 

Fragments of classical industrial cotton (CIC), soft organic cotton (SOC), classical industrial flax (CIF), or soft organic flax (SOF) were layered on *S. aureus* MFP03 (Figure 1A) or *S. epidermidis* MFP04 (Figure 1B) bacterial lawns. Crude or sterilized classical industrial and soft organics cotton showed no effect on *S. aureus* growth. Conversely, a growth inhibition halo was observed around fragments of crude CIF and SOF. The diffusion halo was particularly visible around SOF. Sterilization of the flax fabric led to a complete loss of this activity. Same results were obtained when the fabrics were layered on *S. epidermidis* bacterial lawns. Crude or sterilized cotton were without effect while crude industrial and soft organic flax locally inhibited the bacterial growth. This effect was not observed with sterilized flax fabric.

### 3.2. Effects of Sterilized Cotton and Flax Textile Extracts on Staphylococcus aureus and Staphylococcus epidermidis Growth and Biofilm Formation

Production of fabric extracts requiring incubation in bacterial culture medium (RCM) for 24 h at 37 °C was only possible to realize using sterilized fabric. For that, *S. aureus* MFP03 and *S. epidermidis* MFP04 growth kinetics were studied by culture in undiluted fabric extracts, prepared as described in Materials and Methods Section 2.1, and compared to controls realized using fresh RCM. No significant variation of the generation time was observed between the different growth conditions, and the kinetics were similar (*data not shown*).

The effect of fabrics extracts on bacterial biofilm formation activity of the bacteria was initially studied by a global approach using the crystal violet staining method. Results were expressed as percentages of the basal biofilm values (OD_570nm_ of dye solutions from control biofilms). All fabric extracts showed a marked inhibitory effect on *S. aureus* MFP03 biofilm formation ranging from –47.1 ± 0.1 and −59.7 ± 2.1% with CIC and SOC extracts, respectively, to –74.6 ± 0.9 and –54.9 ± 4.1% with CIF and SOF extracts, respectively (*p* < 0.001) (Figure 2A). Fabric extracts also showed marked effects on *S. epidermidis* MFP04 biofilms formation with inhibitions reaching −65.9 ± 3.6, −71.3 ± 5.2, and—52.6 ± 4.1% for SOC, CIF, and SOF extracts, respectively (Figure 2B). CIC extract had no significant effect on *S. epidermidis* biofilm formation in these experimental conditions.

In order to further investigate the effect of fabric extracts, the biofilms structure was studied by confocal laser scanning microscopy. The maximal biomass and mean thickness of *S. aureus* MFP03 biofilms was increased by CIC extract (+43 ± 8.4 and +63.2 ± 10.7%, respectively, *p* < 0.001), whereas SOC extract induced a marked decrease (−48.5 ± 4.9 and −26.0 ± 1.4%, respectively *p* < 0.001) (Figure 3). The reaction of *S. aureus* MFP03 to flax extracts (CIF and SOF) was characterized by the formation of mushroom-like structures. The mean biomass, mean thickness, and maximal thickness values were then measured on flat areas. In these regions, CIF extracts induced an increase of the mean biomass and thickness (+64.2 ± 5.5 and +66.0 ± 1.7%, respectively *p* < 0.001), whereas SOF extracts led to a moderate decrease of the biomass (−26.7 ± 1.6%, *p* < 0.01) and had no effect on the mean thickness of the biofilms. None of the extracts affected *S. aureus* biofilm maximal thickness values. The response of *S. epidermidis* MFP04 to fabric extracts was homogeneous, and no mushroom-like structures were observed. As shown by the crystal violet straining method, CIC extracts had a limited effect on *S. epidermidis* biofilm structure (Figure 4). An increase in the mean biomass (+21.6 ± 2.6%, *p* < 0.001) was noted, whereas the mean thickness remained unchanged. Only the maximal thickness of the biofilms was marginally reduced (−10.4 ± 0.9%, *p* < 0.001). Conversely, SOC extracts induced a mean reduction of the biomass, mean thickness, and maximal thickness of the biofilms (−54.0 ± 2.3, −60.2 ± 1.4, and −41.7 ± 1.3%, respectively, *p* < 0.001). These results were coherent with data from the crystal violet assay. CIF extracts had a more important impact on *S. epidermidis* biofilm structure with −88.5 ± 0.5, −88.5 ± 1.8, and –57.1 ± 6.1% reduction in the mean biofilm biomass, mean thickness, and maximal thickness, respectively (*p* < 0.001). The effects of SOF extracts were more limited with −33.3 ± 5.3, −38.5 ± 3.7, and −38.4 ± 6.3% reduction of biomass, mean thickness, and maximal thickness, respectively (*p* < 0.001).

Mushroom-like structures formed by *S. aureus* in the presence of flax extracts were studied independently by triple straining using Syto 9 Green, Sypro Ruby, and CalcoFluor in order to visualize bacteria, matrix proteins, and matrix polysaccharides, respectively (Figure 5). Comparison of *x*/*z* transversal reconstructed views revealed that bacteria and the polysaccharides matrix were localized at the basis of the biofilm, whereas protein matrix elements were distributed all over the biofilm thickness and were the essential, if not exclusive, compounds in the mushroom-like structures. This organization was the same in biofilms formed after exposure to CIF or SOF.

### 3.3. Effects of Cotton and Flax Textile Extracts on Staphylococcus aureus and Staphylococcus epidermidis on Biosurfactant Production and Surface Polarity

To obtain further insight into the effects of fabric extracts on biofilm formation, biosurfactant production was investigated by measuring the surface tensions of the RCM medium, containing textile extracts or not, after a 24 h culture with *S. aureus* MFP03 or *S. epidermidis* MFP04 (Figure 6). The RCM medium showed a mean surface tension of 62 mN.m^−1^. All surface tensions of fabric extracts were lower, suggesting that the extraction process introduced some surfactive compounds from the textile. Moreover, except for CIF extracts, the surface tension of fabric extracts was lower than that of both bacterial supernatants. Growing *S. aureus* MFP03 or *S. epidermidis* in the RCM medium did not modify surface tension values. When *S. aureus* MFP03 was grown in CIC extracts, the surface tension of the medium was close to that of CIC extract. In comparison to pure CIC extracts, the surface tension of CIC extracts after *S. epidermidi*s culture was marginally decreased (−10.8 ± 1.0%, *p* < 0.001). A limited decrease of the surface tension of SOC extracts (−5 ± 1.5%, *p* < 0.05) was measured after culture with *S. aureus,* whereas an increase (+12.2 ± 1.3%, *p* < 0.001) was detected after culture of *S. epidermidis* in SOC. The surface tension of CIF extracts was not affected by culture with *S. aureus* or *S. epidermidis*. In the case of SOF extracts, both *S. aureus* and *S. epidermidis* increased the mean surface tension (+17.2 ± 0.9 and +10.4 ± 2.0%, respectively, *p* < 0.001). In all cases, these variations remained marginal, and the surface tension values of fabric extracts exposed to the bacteria never reached the limit of 40 mN.m^−1^ which is considered an indication of biosurfactant production [44,45].

As biofilm formation is linked to bacterial surface properties, the surface polarity and Lewis acid and base properties of *S. aureus* MFP03 and *S. epidermidis* MFP04 grown in control RCM or fabric extracts were measured by the MATS technique (Figure 7). While *S. aureus* MFP03 showed a marked hydrophobic surface (65 ± 3.7% affinity to solvents), that of *S. epidermidis* MFP04 was clearly hydrophilic with extremely limited affinity to solvents (0.3 ± 1.8% affinity to solvents). The two bacteria were also quite different in regard to the Lewis acid and base surface properties. *S. aureus* was moderately basic (28 ± 1.2% Lewis basicity) and acidic (45 ± 0.9% Lewis acidity), whereas the Lewis basicity of *S. epidermidis* was null and its acidity was limited (12 ± 1.2%). None of the fabric extracts had a significant impact on the surface polarity and Lewis acid and base properties of *S. aureus*. A marginal increase of the Lewis basicity was only observed in *S. epidermidis*.

### 3.4. Effect of Cotton and Flax Extracts on Staphylococcus aureus and Staphylococcus epidermidis Resistance to Antibiotics, Cytotoxicity, and Inflammatory Potential

The potential influence of fabric extracts on bacterial resistance to antibiotics was tested over twenty-four antibiotics (Figure 8). The sensitivity of *S. aureus* MFP03 to all antibiotics was partial, with imipenem showing the maximal inhibitory effect. *S. epidermidis* MFP04 also showed partial sensitivity to all tested antibiotics. As previously observed, imipenem but also carbapenem and ticarcillin had maximal inhibitory effects. No difference in sensitivity to antibiotics was noted between bacteria grown in the control RCM medium, CIC, SOC, CIF, or SOF extracts.

The intrinsic cytotoxicity of *S. aureus* MFP03 and *S. epidermidis* MFP04 on HaCaT keratinocytes was moderate with 14.0 ± 1.2 and 6.7 ± 0.52% of maximal cell death measured by LDH assay after exposure to the bacteria (Figure 9). *S. aureus* grown in CIC extract showed a 45.7 ± 8.2% decrease in cytotoxicity in comparison to its mean basal cytotoxicity (*p* < 0.01). The effect of SOC, CIF, and SOF extracts was more pronounced. The cytotoxicity of *S. aureus* was reduced to 8.5 ± 0.9 and 7.4 ± 0.9% of the control after culture in SOC or CIF extracts, respectively (*p* < 0.001). After growth in SOF extracts, the cytotoxicity of *S. aureus* was almost abolished (−99.6 ± 0.05% reduction, *p* < 0.001). The effect of fabric extract on *S. epidermidis* was somehow different, although a net reduction in the cytotoxicity was also observed. CIC, SOC, and SOF extracts led to an almost complete loss of *S. epidermidis* virulence. CIF had a lower impact with only a 66.1 ± 14.0% decrease in the cytotoxicity *(p* < 0.01).

Tests realized using *S. aureus* MFP03 or *S. epidermidis* MFP04 culture supernatants did not show any cytotoxicity on HaCaT cells that should be attributed to the production of soluble toxins.

The effect of fabric extracts on the bacterial inflammatory potential was investigated by assay of IL8 section by HaCaT cells. As previously observed [46], the basal level of IL8 in control conditions was undetectable and no increase of IL8 was measured after exposure to *S. aureus* MFP03 or *S. epidermidis* MFP04 grown in control medium or CIC, SOC, CIF, or SOF fabric extracts *(data not shown).*

### 3.5. Cotton and Flax Extracts Analysis

HPLC analysis of CIC and SOC extracts revealed multiple peaks but did not show any difference between culture conditions (*data not shown*). When flax extracts were analyzed, one peak of higher amplitude was detected in SOF extracts (Figure 10). This peak was identified by gas chromatography coupled to mass spectrometry as benzyl alcohol, an aromatic compound of formula C_6_H_5_CH_2_OH.

## 4. Discussion

Many international standards and regulations including UE 1007/2011 [47] and REACH [48] have been defined to evaluate textiles safety. Specific challenge tests procedures were even recently described for hospital textiles [49]. However, considering their interactions with microorganisms, they are limited to the evaluation of antimicrobial activity, particularly for textiles treated with biocides (DIN EN ISO 20645 2001 [50] and AATCC 147 [51]). These methods have been adapted for assessment of bacterial colonization on untreated fabrics. However, chemicals without effects on bacterial growth can affect their physiology, and particularly virulence [52], and these effects can remain undetectable. Moreover, none of the regulations address the question of the behavior of the microorganisms in their natural cutaneous environment.

In the present study, we focused on two organic textiles, with cotton as the more employed organic fiber all over the world [53] and flax as the more ancient fiber known for its positive effects on skin [54]. Textile samples used in this study were selected as they were not submitted to any post-harvesting treatment. The two model bacteria, *S. aureus* MFP03 and *S. epidermidis* MFP04, were collected on normal skin and their draft genomes were recently described [38]. As shown by direct contact on bacterial lawns, independently of its culture conditions, cotton fabric had no visible antibacterial activity. Conversely, a growth inhibition halo was observed on both staphylococci cultures around crude flax fabrics, suggesting that flax leachables with antimicrobial activity diffused around the samples. This observation is agreement with investigations showing an antibacterial activity of flax fibers characterized by a reduction in the size and number of *S. aureus* colonies on agar plates [55]. The effect of flax extracts was more important with SOF, revealing a potential influence of agricultural practices. Autoclaved samples did not show the same growth inhibitory effect, indicating that the diffusible antimicrobial activity was heat sensitive. However, it is impossible to know if the diffusible compound(s) with antimicrobial activity originated from the flax fiber itself or from its associated microbiota. Fungi residing on flax fibers have shown to secrete enzymes with antibacterial properties [55], but this question requires a qualitative and quantitative analysis of the flax-associated microbiota and deserves a complete separate study.

As we aimed to investigate the effect of textile leachables on cutaneous bacteria, we produced extracts allowing work in vitro on bacterial biofilm formation and virulence. As previously observed, heat treatment abolished flax antimicrobial activity but it was not possible to generate extracts without sterilization and prolonged incubation. Then, we hypothesized that even after autoclaving, extracts should keep sufficient activity on bacterial physiology. In order to interpret results, we tested the effects of extracts on biofilm formation in conditions where they had no effect on bacterial growth kinetics. In agreement with studies realized on textile fragments, none of the textile extracts realized by incubation in the culture medium after sterilization showed growth inhibitory effects.

Using the crystal violet technique, we observed that all tested textile extracts showed inhibitory effect on *S. aureus* MFP03 biofilm formation. Conversely, only SOC, CIF, and SOF extracts inhibited *S. epidermidis* MFP04 biofilms at different levels. CIC displayed no effect in these tests. Altogether, these data showed that although all these textile extracts had no effect on bacterial growth, they contained substance(s) of which some were surfactive, as shown by the decrease in surface tension between extracts and control. These compounds are capable of inhibiting the initial bacterial adhesion and/or biofilm matrix production. In order to investigate the potential effect of textile extracts on biofilms structure, biofilm formation was subsequently studied by confocal laser scanning microscopy. The biomass and mean thicknesses of *S. aureus* biofilms was significantly increased by CIC extracts, whereas it was reduced by SOC extracts. These results are consistent with crystal violet studies, particularly in regard to the inhibitory effect of SOC extracts on *S. aureus* biofilms. The increase observed using CIC extracts may be attributed to the difference in polarity of PVC and glass used as contact surfaces in crystal violet and confocal microscopy studies, respectively. While the former is hydrophobic, the latter is hydrophilic and polar [56]. Nevertheless, these results confirmed the important impact of agricultural practices on textiles properties, suggesting that trace pesticides accumulated in cotton [28] could influence the effect of textiles on cutaneous bacteria. This is to be paralleled to the fact that industrial cotton culture consumes more pesticides and insecticides than any other crop culture [57]. Moreover, most pesticides are non-polar compounds and thus should have high affinity with the *S. aureus* hydrophobic surface [58]. Studies realized on flax have also pointed out the consequences of agricultural practices on the compatibility of textiles with *S. aureus*. CIF increased the mean biomass and thickness of biofilms, whereas SOF decreased its biomass. However, the situation was more complex since exposure of *S. aureus* to flax extracts also led to the formation of mushroom-like structures. As shown by triple labelling, bacteria and polysaccharides appeared localized at the basis of the biofilms, whereas proteins were detected both in flat biofilm areas and in mushroom-like structures. In fact, mushroom-like structures seem to be formed essentially from the proteins-based matrix and should be designated as “empty mushrooms”, as seen in Oprf-protein-deleted *P. aeruginosa* biofilms [59]. As suggested in *P. aeruginosa* [60], mushroom structures should be formed by discrete bacterial subpopulations, and their assembly is regulated by specific genes expression [61]. This is consistent with the difference in sensitivity to flax extracts between bacteria assembled as flat biofilms or mushrooms. Another possibility is that the density of the matrix proteins formed by mushrooms should protect bacteria located at the basis, since matrix permeability can regulate molecules diffusion into biofilm structures [62]. Nevertheless, this formation of mushrooms by *S. aureus,* which can be considered as a defense reaction, was only observed after exposure to CIF and SOF, suggesting that the compound at the origin of this effect is specific to flax and not to flax culture conditions. The impact of textile extracts on *S. epidermidis* MFP04 biofilms observed by confocal microscopy was consistent with crystal violet studies, although a limited increase in mean biomass and decrease in mean thickness was observed with CIC. The effects of textile extracts on *S. epidermidis* were homogeneous and no mushroom-like structures were visualized. Except for CIC, all other textile extracts decreased the mean biomass, mean thickness, and maximal thickness of the biofilms. The maximal effect was observed with CIF extracts.

Biosurfactant production and bacterial surface polarity were determinant parameters in bacterial adhesion and biofilm development [63]. Exposure of bacteria to textile extracts had limited influence on the surface tension values of the growth medium. These values never decreased under the limit of 40 mN.m^−1^, indicating that bacteria were not producing biosurfactant. Similarly, no difference in surface polarity and Lewis acid and base properties were detected between bacteria exposed or not to textile extracts, although the surfaces of *S. aureus* MFP03 and *S. epidermidis* MFP04 have opposite characteristics. The absence of effects of textile extracts on the resistance of bacteria to antibiotics is reassuring, but may also suggest that bacteria had no metabolic reaction to textile extracts and that all effects on biofilm formation were due to differences in diffusion in the biofilm matrix structures. However, this was not the case since we observed that all textile extracts affected the bacterial cytotoxicity. On *S. aureus*, only SOF extracts totally inhibited the cytotoxic effect of the bacteria on HaCaT keratinocytes. As observed in biofilm formation studies, CIC had only a partial inhibitory effect. All textile extracts almost abolished the cytotoxicity of *S. epidermidis* except CIF which had only partial efficiency. The inflammatory response to both bacteria was investigated by IL8 assay, but IL8 levels remained undetectable. Then, these results suggest that the effects of textile extracts on *S. aureus* and *S. epidermidis* biofilm formation were multifactorial and resulted from both bacterial metabolic adaptation and difference of biofilm matrix structure and/or composition.

Identifying all cotton and flax extractables should be a huge work requiring important analytical means. For that reason, we compared HPLC chromatograms between cotton and flax extracts from industrial and soft organic cultures. No differences were observed between cotton extracts from the different origins, suggesting that the technique employed was not sufficient to detect pesticide traces. However, when flax extracts were analyzed, we observed a peak of higher amplitude in extracts from soft organic cultures. This peak was identified as a benzyl alcohol by gas chromatography coupled to mass spectrometry. 

Benzyl alcohol is an aromatic alcohol naturally produced by many plants, and it is frequently found in fruits and tea extracts [64]. It is one of the more commonly employed antimicrobial preservatives [65]. Benzyl alcohol disorganizes the membrane structure and increases the membrane fluidity in Gram-positive and Gram-negative bacteria [66,67]. Flax organic production should apparently favor the presence of this molecule in the final textile. Its presence in flax extracts, under the minimal inhibitory concentration, should be sufficient to affect biofilm formation without acting on bacterial growth and explain, at least in part, the impact on *S. aureus* and *S. epidermidis* of flax extracts. Indeed, as previously shown by Hancock et al. [68], even at sub-lethal concentrations, antimicrobials can affect bacterial physiology, including biofilm formation. 

Considering its inhibitory activity as crude textile on *S. aureus* and *S. epidermidis*, flax showed interesting properties, and particularly so did soft organic cultures that inhibited biofilm development and strongly decreased bacterial cytotoxicity. Soft organic cotton also revealed a valuable potential, although it has no intrinsic antimicrobial activity as crude textile. The effects of cotton and flax extracts on *S. aureus* and *S. epidermidis* were observed after a time of interaction with bacteria (22 to 24 h) exceeding the time that corresponds to normal clothes use. However, these observations should be particularly considered in extreme situations when textiles remain in contact with skin for a long time in confined, hot, and humid environments such as during spaceflights or for people with sensitive skin.

## 5. Conclusions

Taken together, this study revealed that crude flax fibers can inhibit *S. aureus* and *S. epidermidis* development and that cotton and flax sterile extract can modulate biofilm formation by these bacteria. Flax extracts, and particularly those produced by organic agricultural practices, were particularly active, suggesting that even in the absence of antimicrobial treatment, textile leachables can affect the physiology of cutaneous bacteria. The impact of other types of textiles requires further evaluations.

## Figures and Tables

**Figure 1 life-12-00535-f001:**
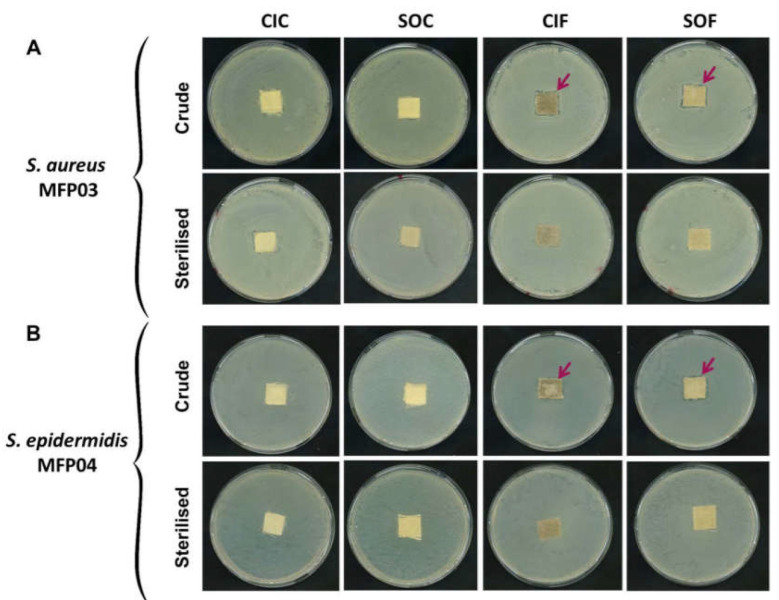
Effects of crude or sterilized cotton or flax textile samples produced by classical industrial or soft organic agricultural practices on (**A**) *Staphylococcus aureus* MFP03 and **(B**) *Staphylococcus epidermidis* MFP04 lawns on solid medium. CIC: classical industrial cotton, SOC: soft organic cotton, CIF: classical industrial flax, SOF: soft organic flax. Arrows indicate growth inhibition areas.

**Figure 2 life-12-00535-f002:**
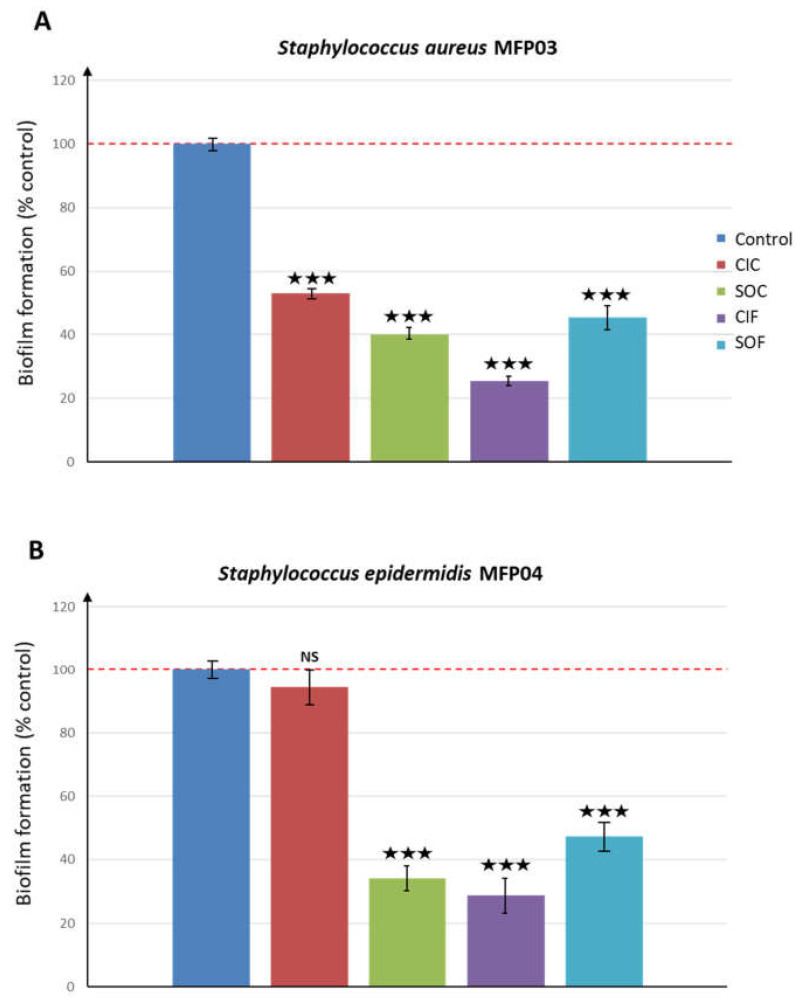
Effects of classical industrial cotton (CIC), soft organic cotton (SOC), classical industrial flax (CIF), and soft organic flax (SOF) extracts on (**A**) *Staphylococcus aureus* MFP03 and (**B**) *Staphylococcus epidermidis* MFP04 biofilm formation studied by the crystal violet technique on flat-bottom polystyrene plates. Results are representative of three independent experiments. (NS: not significant; ★★★ = *p* < 0.001).

**Figure 3 life-12-00535-f003:**
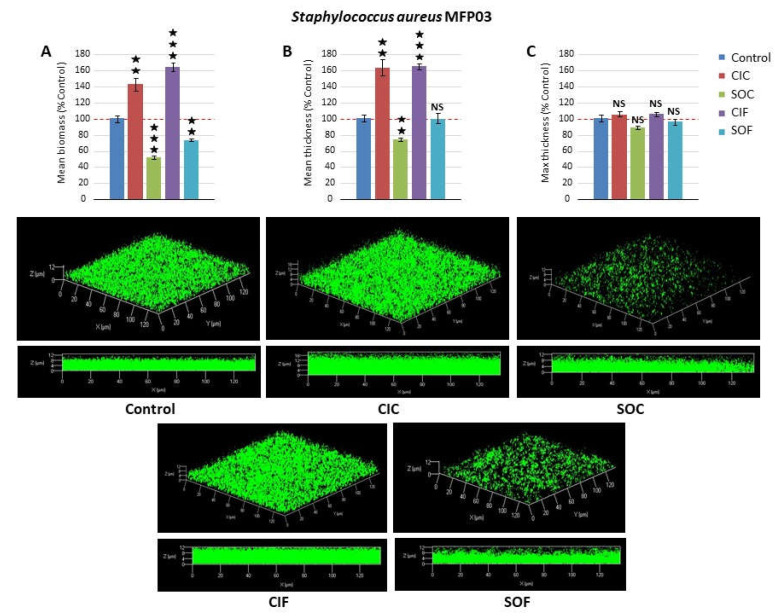
Effects of classical industrial cotton (CIC), soft organic cotton (SOC), classical industrial flax (CIF), and soft organic flax (SOF) extracts on *Staphylococcus aureus* MFP03 biofilm formation studied by confocal laser scanning microscopy. Figure’s top views show (**A**) the calculated mean biofilm biomass, (**B**) mean thickness, and (**C**) maximal thickness. Images (*x/y* top views and *x/z* lateral views) of representative biofilms formed in the absence or presence of textile extracts are shown in lower cases. (NS: not significant; ★★ = *p* < 0.01, ★★★ = *p* < 0.001).

**Figure 4 life-12-00535-f004:**
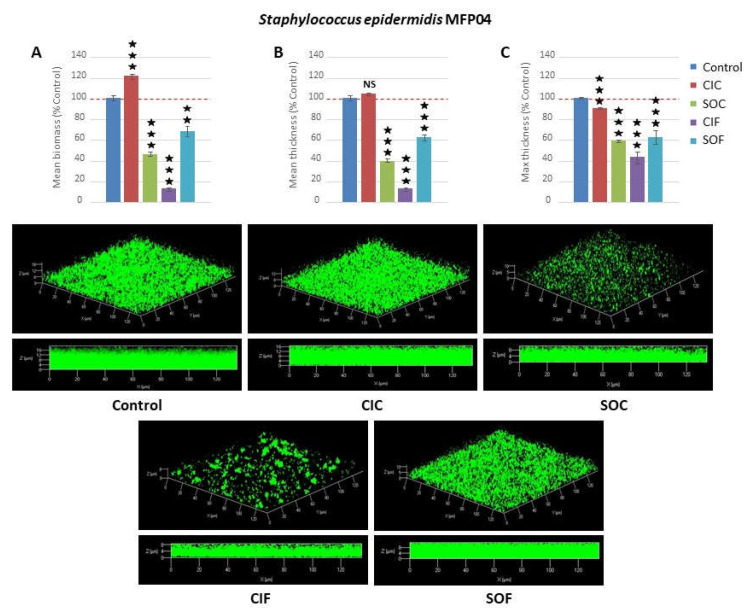
Effects of classical industrial cotton (CIC), soft organic cotton (SOC), classical industrial flax (CIF), and soft organic flax (SOF) extracts on *Staphylococcus epidermidis* MFP04 biofilm formation studied by confocal laser scanning microscopy. Figure’s top views show (**A**) the calculated mean biofilm biomass, (**B**) mean thickness, and (**C**) maximal thickness. Images (*x/y* top views and *x/z* lateral views) of representative biofilms formed in the absence or presence of textile extracts are shown in lower cases. (NS: not significant; ★★ = *p* < 0.01, ★★★ = *p* < 0.001).

**Figure 5 life-12-00535-f005:**
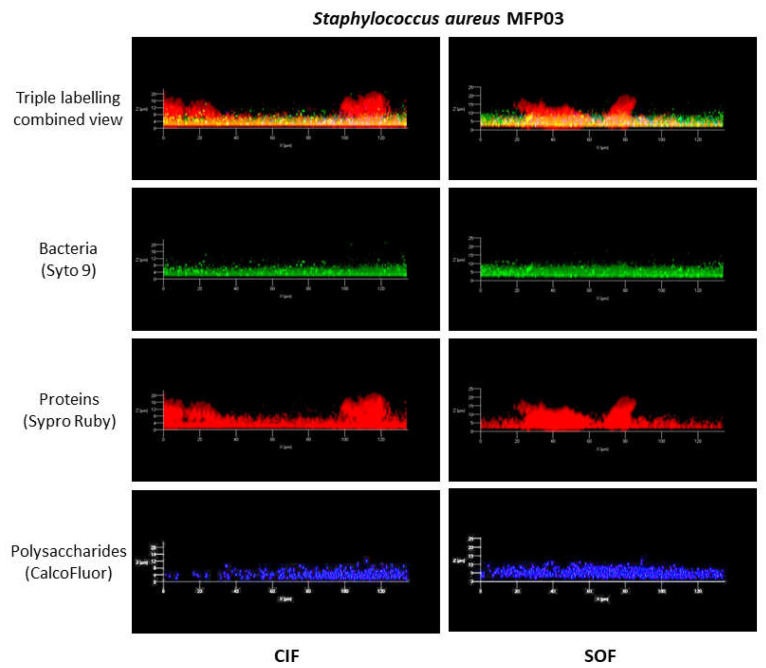
Visualization of mushroom-like structures formed by *S. aureus* in the presence of classical industrial flax (CIF) and soft organic flax (SOF) extracts. Biofilms were studied by triple straining using Syto 9 (green), Sypro Ruby (red), and CalcoFluor (blue) labelling bacteria, matrix proteins, and matrix polysaccharide, respectively.

**Figure 6 life-12-00535-f006:**
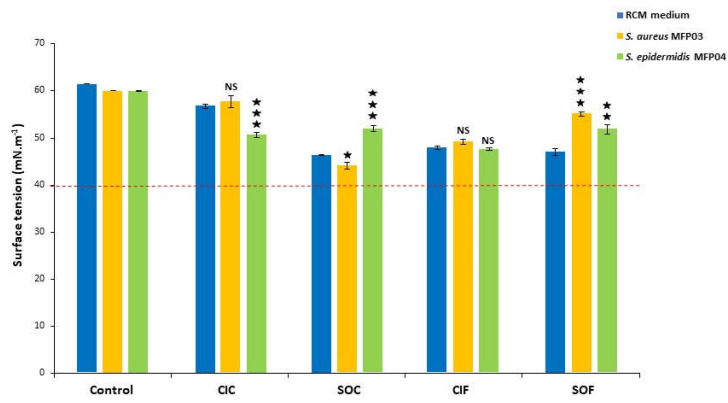
Surface tension of the control RCM medium, classical industrial cotton (CIC), soft organic cotton (SOC), classical industrial flax (CIF), and soft organic flax (SOF) extracts in RCM and same solutions after growth of *Staphylococcus aureus* MFP03 or *Staphylococcus epidermidis* MFP04 for 24 h. The red line shows the value of 40 mN.m^−1^ which is considered as the limit indicative of the presence of biosurfactant in the medium. Differences were calculated in regard to the corresponding medium (NS: not significant; ★ = *p* < 0.05; ★★ = *p* < 0.01; ★★★ = *p* < 0.001).

**Figure 7 life-12-00535-f007:**
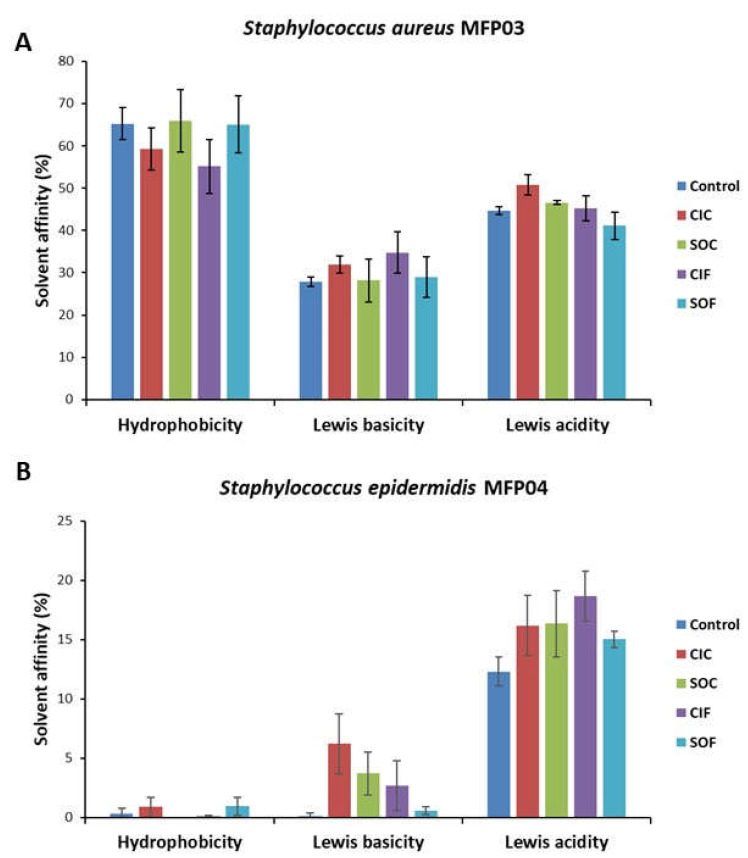
Effect of classical industrial cotton (CIC), soft organic cotton (SOC), classical industrial flax (CIF), and soft organic flax (SOF) extracts on the surface hydrophobicity, Lewis basicity, and Lewis acidity of *Staphylococcus aureus* MFP03 (**A**) or *Staphylococcus epidermidis* MFP04 (**B**) determined by the microbial adhesion to solvents (MATS) technique.

**Figure 8 life-12-00535-f008:**
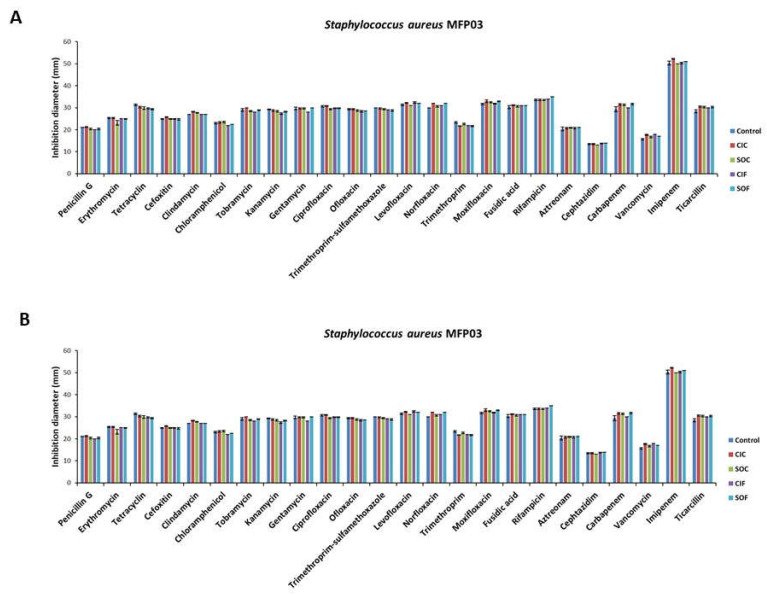
Evaluation of *Staphylococcus aureus* MFP03 (**A**) and *Staphylococcus epidermidis* MFP04 (**B**) resistance to 24 antibiotics after growth in RCM medium, classical industrial cotton (CIC), soft organic cotton (SOC), classical industrial flax (CIF), or soft organic flax (SOF) extracts in RCM using the disk diffusion method.

**Figure 9 life-12-00535-f009:**
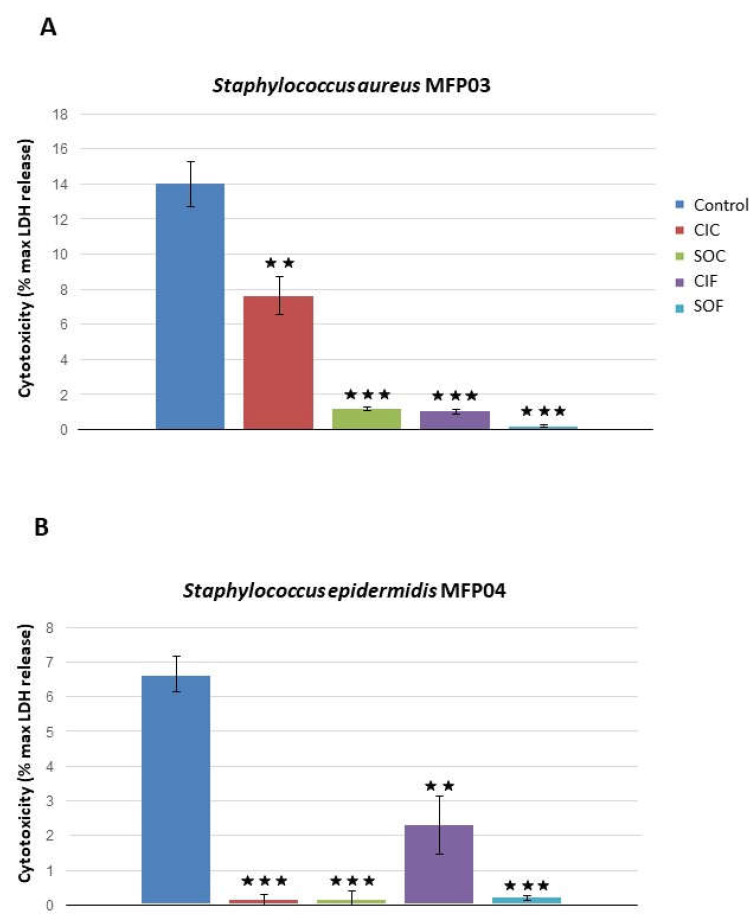
Effect of classical industrial cotton (CIC), soft organic cotton (SOC), classical industrial flax (CIF), and soft organic flax (SOF) extracts on *Staphylococcus aureus* MFP03 (**A**) and *Staphylococcus epidermidis* MFP04 (**B**) cytotoxicity on HaCat cells measured by the lactate dehydrogenase (LDH) release assay. (★★ = *p* < 0.01; ★★★ = *p* < 0.001).

**Figure 10 life-12-00535-f010:**
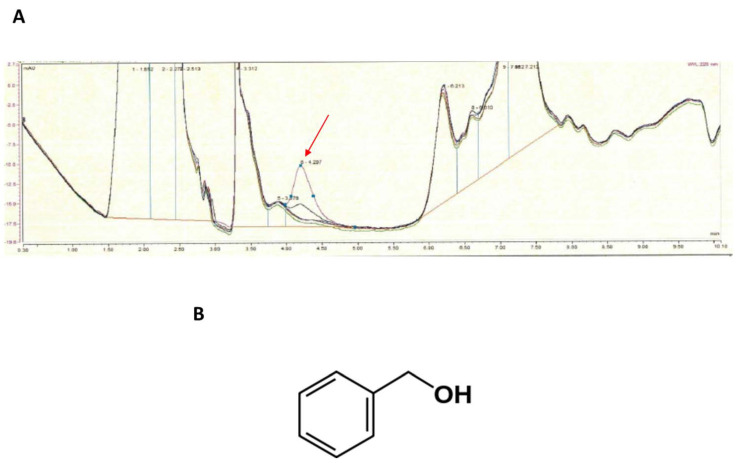
Comparison of chromatogram of classical industrial flax (CIF) and soft organic flax (SOF) extracts after separation by HPLC, indicating the presence of a specific peak (arrow) that was only observed in SOF extracts (**A**). Structure of benzyl alcohol identified by GC-ESI MS as the molecule present in this peak (**B**).

**Table 1 life-12-00535-t001:** Antibiotics used for testing bacterial resistance and respective tested concentrations according to EUCAST protocol.

Antibiotics	Concentrations (µg)
Penicillin G	1
Erythromycin	15
Tetracyclin	30
Cefoxitin	30
Clindamycin	2
Chloramphenicol	30
Tobramycin	10
Kanamycin	30
Gentamycin	10
Ciprofloxacin	5
Ofloxacin	5
Trimethroprim-sulfamethoxazole	25
Levofloxacin	5
Norfloxacin	10
Trimethroprim	5
Moxifloxacin	5
Fusidic acid	10
Rifampicin	5
Aztreonam	15
Cephtazidim	10
Carbapenem	100
Vancomycin	30
Imipenem	10
Ticarcillin	75

## Data Availability

All data presented in this study are available on request from the corresponding author.

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
