# Peer review of "Cotton and Flax Textiles Leachables Impact Differently Cutaneous Staphylococcus aureus and Staphylococcus epidermidis Biofilm Formation and Cytotoxicity"

_life, 2022, doi:10.3390/life12040535_

Round 1

Reviewer 1 Report

In this manuscript the properties of cotton and flax textile leachables are evaluated.

on two types of Staphylococcus, especially the formation of biofilm and the cytotoxic effect.

The following comments are made:

  1. Lines 88-89. What textiles do you refer to: CIC, CIF, SOC, SOF? You can clear it up.
  2. Line 110. How to spread them?
  3. Line 125. Where were they collected from?
  4. Line 127. How many CFUs?
  5. Line 138. What scale did you use for biofilm formation?
  6. Line 191. How many CFUs?
  7. Lines 191- 192. How did you measure the number of bacteria?
  8. Line 206. What antibiotics were used and what concentrations?
  9. Line 224. What does LDH mean?
  10. Lines 283-284, and 286. What the values ​​mean. It is not clear. How did you calculate the values?
  11. Figure 3. What is the difference in the top three graphs? Indicate it in the foot of the figure with letters.
  12. Figure 4. Same observation as Figure 3.
  13. Figure 3. “ *= p < 0.05; **= p < 0.01)” In the Figure there are ** and *** to correct
  14. Figure 4: “***= p < 0.001)”. In the figure there are ** and *** correct

15.Line 482-483. Where are the crystal violet biofilm measurement data in the results section?

  1. Discuss your results in relation to the time the textiles are in contact with the skin.
  2. Also discuss what type of textiles has better activity and why

Author Response

Point by point responses to Reviewer 1:

  1. CIC, SOF; SOC and SOF have been more precisely described in subchapter 2.1
  2. The way bacteria were homogeneously spread on agar surfaces has been detailed in  subchapter 2.3.
  3. Production of bacteria for biofilm formation studies is now indicated in subchapter 2.5.
  4. Correspondence between OD580nm and CFU/mL is now indicated in subchapter 2.5.
  5. As mentioned, biofilm formation is expressed as percentages of the control values. The linearity of the OD detection was indicated in subchapter 2.5.
  6.  Correspondence between OD580nm and CFU/mL is now indicated in subchapter 2.8.
  7. Measurement of bacterial affinity to solvents was described with more details in subchapter 2.8.
  8. Tested antibiotics and their respective concentrations are now presented in Table 1 as indicated in subchapter 2.9.
  9. The complete name of LDH is now given on subchapter 2.10.
  10. It is now indicated in subchapter 3.2 that results of cristal violet biofilm formations were expressed as percentages of the basal biofilm values. The reference of the technique is now indicated (reference 39).
  11. Figure 3 and its caption were modified including references to graphs A, B; C in top of the figure.
  12. Figure 4 and its caption were modified including references to graphs A, B; C in top of the figure.
  13. Figure 3 caption was modified in respect to the p values in the graphs.
  14. Figure 4 caption was modified in respect to the p values in the graphs.
  15. Crystal violet biofilms results are presented in the results subchapter 3.2 and in Figure 2.
  16. A new part was added at the end of the discussion chapter to discuss textiles interactions with skin and relative activities.

Reviewer 2 Report

The manuscript titled ‘Cotton and flax textiles leachables impact differently cutaneous Staphylococcus aureus and Staphylococcus epidermidis biofilm formation and cytotoxicity’ investigates the impact of cotton and flax produced from classical and soft ecological agriculture -i.e. Classical industrial cotton (CIC), soft organic cotton (SOC), classical industrial flax (CIF) and soft organic flax (SOF) textiles - on S. aureus and S. epidermidis, as  direct effect of textiles and, also, as extracts produced with the purpose to investigate the potential of textiles leachables to act on various properties of bacteria; paramenters such the growth curves, biofilm formation activity, biosurfactant production, surface properties, antibiotic resistance, cytotoxicity and inflammatory potential of aforementioned bacteria were investigated.  The research is sound, adds some elements of novelty; the manuscript is thoroughly documented and the methodology is adequately chosen; the perspective is somewhat fresh and updated. The conclusions are supported by the results.

Introduction section could be expanded, as to incorporate more information on skin microbiota.

A part of the Conclusions section could be moved into the Discussion section where if fits better (‘This should be particularly considered in extreme situations, when textiles remain in contact with skin in confined, hot and humid environments, such as in spaceships, or for sensitive skin people’).

Grammar and punctuation must also be carefully checked within the entire article  (e.g. ‘heath-treatment../diffusible antimicrobial activity was heath sensitive’..)

Author Response

Point by point responses to Reviewer 2:

            Three sentences and 2 new references were added in the introduction chapter in order to present the cutaneous skin microbiota.

            The conclusion was modified and the point concerning the interaction of skin with textiles in extreme situations was moved to the last part of the discussion chapter.

            We tried to remove grammatical errors and to introduce necessary punctuations. However, none of us is a native English speaker, and time for return of the revised version was too short for having a reviewing by a professional service.

Round 2

Reviewer 1 Report

1. Line 306, Figure 2. No column has *p, so don't set it.
2. Figure 3 and its caption were modified including references to graphs A, B; C in top of the figure. It was not done.
3. Figure 4 and its caption were modified including references to graphs A, B; C in top of the figure. It was not done.
4. The numbering of the references appears to be a subscript, put the normal number.

Author Response

  1. The *p in Figure 2 caption was suppressed
  2. References to graphs A, B and C have been more clearly indicated in bold in Figure 3 caption.
  3. References to graphs A, B and C have been now indicated in bold in Figure 4 caption.
  4. We apologize, but in our original manuscript, the numbering of the references was not in subscript. This change only appeared after conversion to the journal format by the submission system.

Round 3

Reviewer 1 Report

The authors made the suggested changes